# Restoration of MPTP-induced deficits by exercise and Milmed® co-treatment

Trevor Archer[1,2], Danilo Garcia[2,3] and Anders Fredriksson[4]

[1] Department of Psychology, University of Gothenburg, Gothenburg, Sweden
[2] Network for Empowerment and Well-Being, Sweden
[3] Institute of Neuroscience and Physiology, Centre for Ethics, Law and Mental Health (CELAM), University of Gothenburg, Gothenburg, Sweden
[4] Department of Neuroscience Psychiatry, Uppsala University, Uppsala, Sweden

## ABSTRACT

1-methyl-4-phenyl-1,2,3,6-tetrahydropyridine (MPTP) induces permanent neurochemical and functional deficits. Following the administration of either two or four injections of the dopamine neurotoxin, MPTP, at a dose of 40 mg/kg, C57/BL6 mice were given access to running-wheels (30-min sessions, four times/week, Monday–Thursday) and treatment with the treated yeast, Milmed® (four times/week, Monday–Thursday), or simply running-wheel exercise by itself, over ten weeks. It was observed that the combination of physical exercise and Milmed® treatment, the MPTP + Exercise + Yeast (MC) group [MPTP + Exercise + Milmed® (MC)], restored spontaneous motor activity markedly by test day 10, restored completely subthreshold L-Dopa-induced activity, and dopamine concentration to 76% of control values, in the condition wherein two administrations of MPTP (2 × 40 mg/kg) were given prior to initiation of exercise and/or Milmed® treatment. Physical exercise by itself, MPTP + Exercise (MC) group, attenuated these deficits only partially. Administration of MPTP four times (i.e., 40 mg/kg, s.c., once weekly over four weeks for a total of 160 mg/kg, MPTP + Exercise + Yeast (MC) group [MPTP + Exercise + Milmed® (SC)] and MPTP + Exercise (SC), induced a lesioning effect that was far too severe for either exercise alone or the exercise + Milmed® combination to ameliorate. Nevertheless, these findings indicate a powerful effect of physical exercise reinforced by Milmed® treatment in restoring MPTP-induced deficits of motor function and dopamine neurochemistry in mice.

Although physical exercise by itself ameliorates functional, motor activity, and neurochemical, dopamine, deficits induced by the selective dopamine neurotoxin, 1-methyl-4-phenyl-1,2,3,6-tetrahydropyridine (MPTP, *Archer & Fredriksson, 2010*; *Archer & Fredriksson, 2012*; *Archer & Fredriksson, 2013a*; *Archer & Fredriksson, 2013b*; *Archer & Fredriksson, 2013c*; *Fredriksson et al., 2011*), the combination of physical exercise with Milmed® provided evidence of complete restoration of these deficits (*Archer & Fredriksson, 2013a*). The efficacy of physical exercise (*Archer, 2011*; *Archer, Johansson & Fredriksson, 2011*; *Earhart & Falvo, 2013*) and physiotherapy (*Cholewa, Boczarska-Jedynak*

Corresponding author
Danilo Garcia,
danilo.garcia@neuro.gu.se,
danilo.garcia@euromail.se

& *Opala, 2013*) for Parkinsonism and other neurodegenerative conditions has been well-documented with ever-increasing evidence from both clinical and laboratory studies (*Park et al., 2014*; *Wang et al., 2013*; *Zigmond & Smeyne, 2014*). Three main conclusions were obtained from a recent Milmed[®] study (*Archer & Fredriksson, 2013a*): (i) dopamine integrity was observed to be a direct function of ability to express running exercise in a treadmill running arrangement, (ii) dopamine integrity was observed to be a direct function of the capacity for motor performance as measured by spontaneous motor activity and subthreshold L-Dopa (5 ml/kg) induced activity in the motor activity test chambers, and (iii) running exercise in the treadmill running wheel predicted later motor performance in the motor activity test chambers to an extremely high degree. In this respect, it has been demonstrated in rats that caffeine, both hyperthermic and ergogenic, in combination with physical exercise increased extracellular dopamine and noradrenaline in the preoptic area and anterior hypothalamus (*Zheng et al., 2014*). Treadmill exercise ameliorated also the nigrostriatal dopaminergic neuronal loss in adolescent rats following neonatal hypoxic brain ischemia which improved spatial learning ability (*Park et al., 2013*). Finally, using the $10 \times 25$ mg/kg MPTP model of Parkinson's disease it was indicated that four weeks of treadmill running decreased the levels of the inducible form of nitric oxide and neuronal nitric oxide in the brains of MPTP-treated mice (*Al-Jarrah, 2013*).

The preparation and application of treated yeast culture, *Saccharoyces cerevisiae*, to provide the antiparkinson agent, Milmed[®], in suspension form has been outlined previously (*Gedymin et al., 1999*; *Kolosova et al., 1998*). The derivation of Milmed[®] for cellular genesis and reparation has been reported elsewhere (*Golant, 1994*; *Golant et al., 1994*; *Ragimov et al., 1991*). The description, preparation, and application of the Milmed[®] suspension (oral administration, four times a week) in an animal MPTP model of Parkinsonism has also been described comprehensible in earlier studies (*Archer & Fredriksson, 2013a*; *Archer & Fredriksson, 2013b*). Generally, studies designed to apply exercise intervention using MPTP lesioning to induce Parkinson symptoms, e.g., hypokinesia, in the laboratory have introduced exercise, with or without (*Archer & Fredriksson, 2010*) Milmed[®] co-treatment, prior to administration of the neurotoxin, was shown to prevent any MPTP-induced deficits in several experiments. In the present study, C57/BL6 mice were administered MPTP (40 mg/kg) either twice or four times before access to the exercise (running-wheel, four 30 min/sessions/week, Mon.–Thurs.) and Milmed[®] co-treatment intervention (once/day, four times/week). The purpose of the present study was to ascertain the extent to which the exercise + Milmed[®] treatment combination would restore MPTP-induced functional and neurochemical deficits following either two (moderate condition; [MPTP + Exercise + Milmed[®] (MC)]) or four (severe condition; [MPTP + Exercise + Milmed[®] (SC)]) injections of the neurotoxin.

## MATERIALS AND METHODS

### Ethics statement

The study was carried out in accordance with the European Communities Council Directive of 24th November 1986 (86/609/EEC) after approval from the local ethical Committee

(Uppsala University and Agricultural Research Council), and by the Swedish Committee for Ethical Experiments on Laboratory animals (License S93/92 and S77/94, Stockholm, Sweden). Throughout all the series of experiments performed with Milmed® there is no evidence of any intestinal problem or any other side effects. Additionally, Milmed® has been administered, orally, to more than 1,000 human subjects for the alleviation of several ailments, including Parkinson's disease, borellia, cancer, and attention-deficit hyperactive disorder, as well as a nutritional adjunct for elite-level athletes, over periods extending from eight-to-sixteen weeks; no side effects have ever been reported.

## Animals

Male C57/BL6 mice, purchased from B&K, Sollentuna, Sweden, maintained five-to-ten in a cage in plastic cages in an isolated room at $22 \pm 1\,°C$ and 12 h/12 h constant light/dark cycle (lights on between 06.00 and 18.00 h), were acclimatized, housed and given access to the running-wheels or holding-cages (30-min sessions) in an identical to that described previously (*Archer & Fredriksson, 2013a*). Each group consisted of 10 mice.

## Drugs

MPTP (Research Biochemical Inc., MA, USA), injected $4 \times 40$ mg/kg, s.c. (1-week intervals between injections, was dissolved in saline and administered in a volume of 2 ml/kg body weight. Milmed® was prepared for administration according to a procedure identical to that described previously (*Archer & Fredriksson, 2013a*). Mice were administered oral injections of 0.5 ml/kg Milmed® containing a cell concentration of approximately $2 \times 10^6$ yeast cells daily, according to the preparation protocol and design developed from previous observations regarding stability and viability of the compound (each dose contained $1 \times 10^6$ yeast cells). Each mouse was administered Milmed® once each day four times/week during the 10 weeks of exercise + Milmed® treatment with Design and treatment maintained as described previously (cf. *Archer & Fredriksson, 2013a*).

## Behavioural measurements and apparatus

Testing of motor activity in the ADEA test chambers where Locomotion, Rearing and Total activity were measured was performed in an identical manner to that described previously (*Archer et al., 1986*). Activity test chambers: An automated device, consisting of macrolon rodent test cages ($40 \times 25 \times 15$ cm) each placed within two series of infra-red beams (at two different heights, one low and one high, 2 and 8 cm, respectively, above the surface of the sawdust, 1 cm deep), was used to measure spontaneous motor activity (RAT-O-MATIC; ADEA Elektronic AB, Uppsala, Sweden). The distance between the infra-red beams was as follows: the low levels beams were 73 mm apart lengthwise and 58 mm apart breadthwise in relation to the test chamber; the high level beams, placed only along each long side of the test chamber were 28 mm apart. According to the procedures described previously (*Archer et al., 1986*), the following parameters were measured: LOCOMOTION was measured by the low grid of infra-red beams. Counts were registered only when the mouse in the horizontal plane, ambulated around the test-cage. REARING was registered throughout the time when at least one high level beam was interrupted, i.e., the number of counts

registered was proportional to the amount of time spent rearing. TOTAL ACTIVITY was measured by a sensor (a pick-up similar to a gramophone needle, mounted on a lever with a counterweight) with which the test cage was constantly in contact. The sensor registered all types of vibration received from the test cage, such as those produced both by locomotion and rearing as well as shaking, tremors, scratching and grooming. All three behavioural parameters were measured over three consecutive 20-min periods. The motor activity test room, in which all 12 ADEA activity test chambers, each identical to the home cage, were placed, was well-secluded and used only for this purpose. Each test chamber (i.e., activity cage) was placed in a sound-proofed wooden box with 12 cm thick walls and front panels, and day-lighting. Motor activity parameters were tested on one occasion only, over three consecutive 20-min periods, at the age of three to four months.

Running-wheel units. These were small rodent running exercise wheels, purchased from a Pet store and considered suitable for small rodents. The wheels were adapted and modified for use by mice and placed altogether in a large sound-proofed room within the animal section of the laboratory. All 25 running-wheels were placed equidistant from each other with adjacent wheels in two long rows such that the sounds of the wheels turning by any one wheel could easily be heard by the occupants of all the other wheels. A photograph of the types of running wheel used, presenting a row of the activity running-wheels applied in all the experiments as well as the 'holding' cages in which the non-exercise groups remained, is depicted previously (*Archer & Fredriksson, 2010*). In previous neuroteratological studies that observed wheel-running exercise following different types of perinatal treatments it was observed that each wheel had to be isolated from each of the others since the noise emitted by one animal served to evoke wheel-running behavior in the other animals. However, for the purposes of the present experiments it was considered to be an advantage if the mice in the Exercise groups stimulated each other to perform physical exercise. Access to the running-wheels over daily 30-min sessions was maintained as described previously (*Archer & Fredriksson, 2010*).

## Procedure

The identical procedure to that employed previously (*Archer & Fredriksson, 2012*) was maintained. Access to the running wheel was presented on the 1st four days of the week (Mon.–Thurs.) and motor activity testing on the 5th day (Friday), as previously (*Archer & Fredriksson, 2010*). Testing consisted of spontaneous motor activity test (60 min) and L-Dopa-induced activity test (180-min).

## Neurochemical analysis

Analysis of striatal dopamine concentrations was performed in an identical manner to that described previously (*Archer & Fredriksson, 2013a*). Mice were scarified by cervical dislocation within two weeks of completion of behavioral testing. Determination was carried out using a high-performance liquid chromatograph with electrochemical detection, according to *Björk et al. (1991)*, with modifications (*Liu et al., 1995*). The dopamine analysis concentration results are expressed as ng/ml wet weight of tissue.

### Statistical analysis

Spontaneous motor activity counts (60-min test sessions) and L-Dopa-induced motor activity counts (180-min test sessions) were subjected Split-plot ANOVA whereas striatal dopamine concentrations were subjected to one-way ANOVA (*Kirk, 1995*). Pairwise testing between groups was performed using Tukey's HSD tests.

## RESULTS

The combination of physical exercise with Milmed® restored both function, spontaneous motor activity and L-Dopa-induced activity, and neurochemical, dopamine, deficits whereas exercise, by itself, attenuated the deficits.

### Spontaneous motor activity

Split-plot ANOVA indicated a significant Groups x Test days interaction for locomotion, rearing and total activity counts: $F(41, 419) = 97.61$, $p < 0.0001$; $F(41, 419) = 69.23$, $p < 0.0001$; and, $F(41, 419) = 50.13$, $p < 0.0001$, respectively. Figure 1 presents the mean and standard deviation (*SD*) for locomotion, rearing and total activity counts for each of the six groups: Vehicle, MPTP, MPTP + Exercise (MC), MPTP + Exercise + Yeast (MC) [MPTP + Exercise + Milmed® (MC)], MPTP + Exercise (SC), and MPTP + Exercise + Yeast (SC) [MPTP + Exercise + Milmed® (SC)], over 60-min test sessions for the spontaneous activity tests.

  Pairwise testing using *Tukey's HSD* test indicated that over all three motor activity parameters that: (i) the MPTP + Exercise (MC) and MPTP + Exercise + Yeast (MC) groups made more counts than the MPTP, MPTP + Exercise (SC) and MPTP + Exercise + Yeast (SC) [MPTP + Exercise + Milmed® (SC)] groups but fewer counts than the Vehicle group, (ii) the MPTP + Exercise + Yeast (MC) [MPTP + Exercise + Milmed® (MC)] group made more counts than the MPTP + Exercise (MC) group over test days 6, 8, 10, and (iii) the MPTP + Exercise (MC) and MPTP + Exercise + Yeast (MC) groups increased the numbers of counts made from test days 3–10 days, an indication of gradual recovery.

### L-Dopa-induced activity

Split-plot ANOVA indicated a significant Groups x Test days interaction for locomotion, rearing and total activity counts: $F(17, 143) = 72.81$, $p < 0.0001$; $F(17, 143) = 15.81$, $p < 0.0001$; and $F(17, 143) = 15.97$, respectively. Figure 2 presents the mean and SD for locomotion, rearing and total activity counts for each of the six groups: Vehicle, MPTP, MPTP + Exercise (MC), MPTP + Exercise + Yeast (MC) [MPTP + Exercise + Milmed® (MC)], MPTP + Exercise (SC), and MPTP + Exercise + Yeast (SC) [MPTP + Exercise + Milmed® (SC)], over 180-min test sessions for the L-Dopa-induced activity test.

  Pairwise testing using *Tukey's HSD* test indicated that over all three motor activity parameters that: (i) the MPTP + Exercise + Yeast (MC) group made more counts than all the other MPTP-injected groups and as many counts as the vehicle group, (ii) the MPTP + Exercise (MC) [MPTP + Exercise + Milmed® (MC)] group made more counts than the the MPTP, MPTP + Exercise (SC) and MPTP + Exercise + Yeast

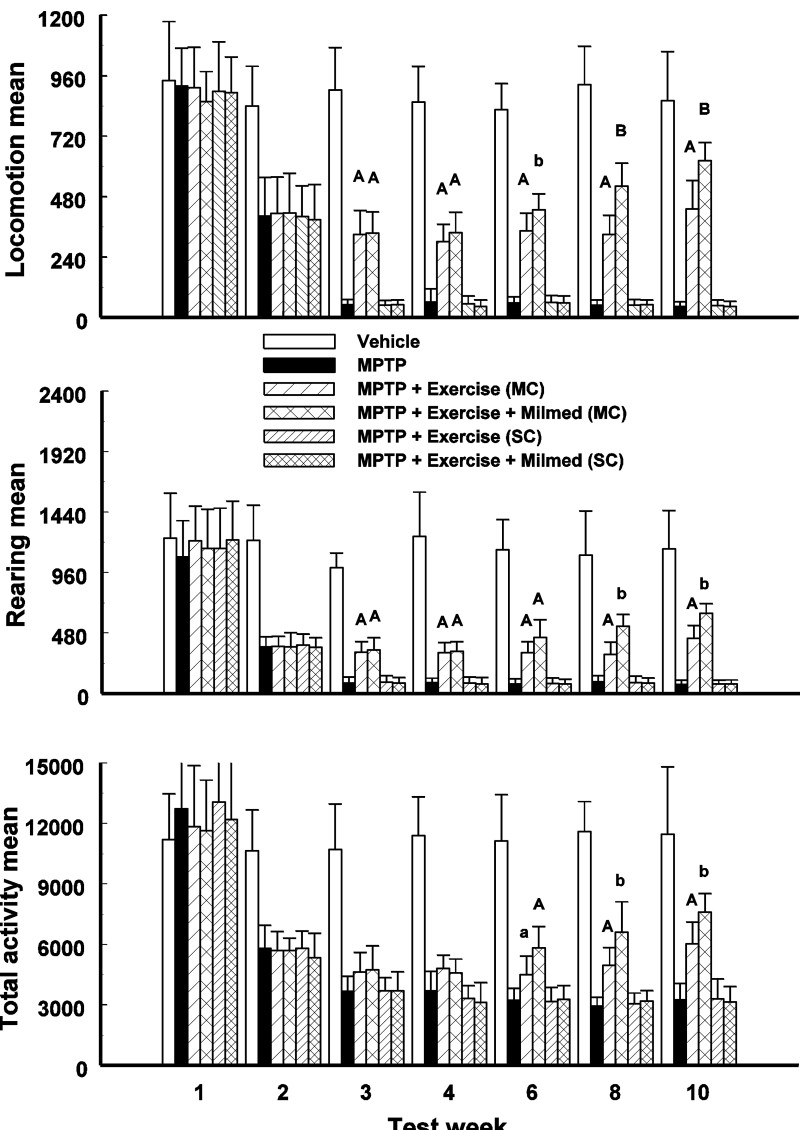

**Figure 1 Spontaneous motor activity.** Mean (SD) locomotion, rearing and total activity counts for each of the six groups: Vehicle, MPTP, MPTP + Exercise (MC), MPTP + Exercise + Yeast (MC) [MPTP + Exercise + Milmed® (MC)], MPTP + Exercise (SC), and MPTP + Exercise + Yeast (SC) [MPTP + Exercise + Milmed® (SC)], over 60-min test sessions for the spontaneous activity tests. MPTP was injected (40 mg/kg, s.c., single weekly injections) either twice or four times prior to initiation of wheel-running exercise (30-min sessions/week, Mon.–Thurs.) + Milmed treatment (four injections, p.o., each week). Pairwise comparisons: A versus MPTP group, $p < 0.01$, a versus MPTP group, $p < 0.05$; B versus MPTP + Exercise (MC) group, $p < 0.01$ b versus MPTP + Exercise (MC) group, $p < 0.05$.

(SC) [MPTP + Exercise + Milmed® (SC)] groups, and (iii) the motor activity of the MPTP + Exercise + Yeast (MC) group was restored completely.

## Neurochemical analysis

One-way ANOVA indicated a significant between-groups effect for striatal dopamine concentrations: $F(5, 30) = 55.53$, $p < 0.0001$. Figure 3 presents the mean and *SD* in

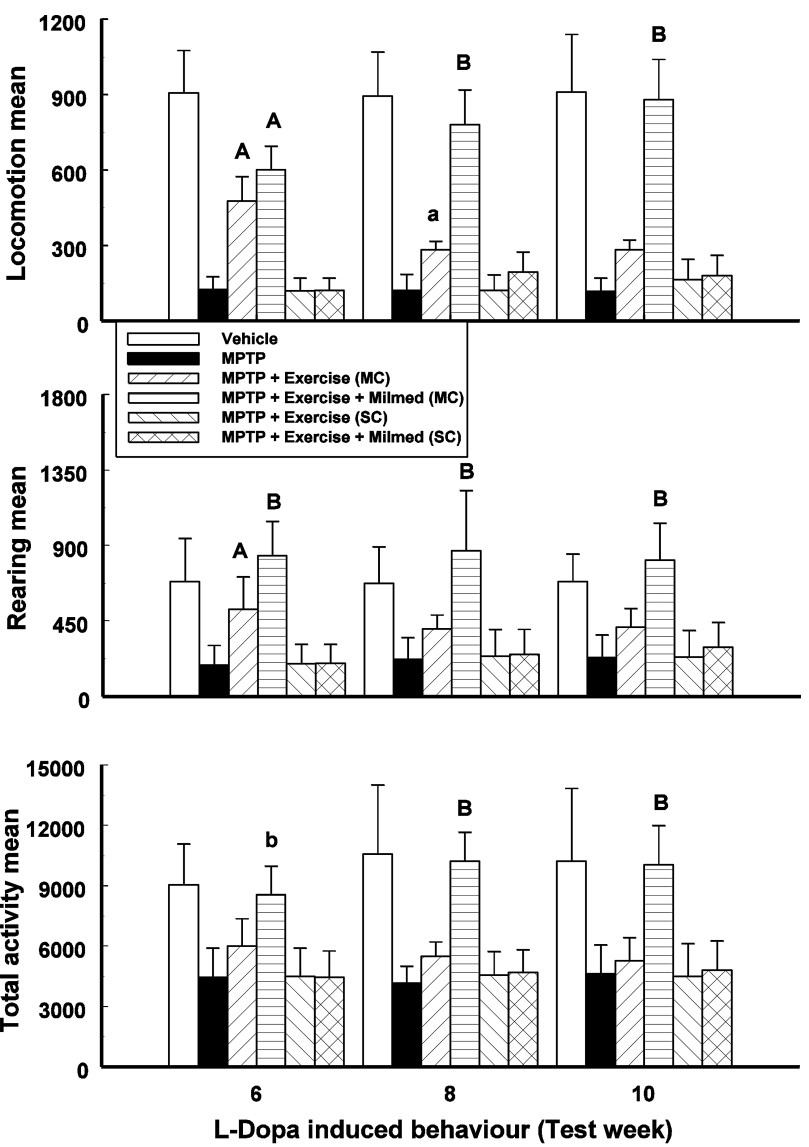

**Figure 2 L-Dopa-induced activity.** Mean (SD) locomotion, rearing and total activity counts for each of the six groups: Vehicle, MPTP, MPTP + Exercise (MC), MPTP + Exercise + Yeast (MC) [MPTP + Exercise + Milmed® (MC)], MPTP + Exercise (SC), and MPTP + Exercise + Yeast (SC) [MPTP + Exercise + Milmed® (SC)], over 180-min test sessions for the L-Dopa-induced activity tests. MPTP was injected (40 mg/kg, s.c., single weekly injections) either twice or four times prior to initiation of wheel-running exercise (30-min sessions/week, Mon.–Thurs.) + Milmed treatment (four injections, p.o., each week). Pairwise comparisons: A versus MPTP group, $p < 0.01$, a versus MPTP group, $p < 0.05$; B versus MPTP + Exercise (MC) group, $p < 0.01$.

dopamine concentrations for each of the six groups: Vehicle, MPTP, MPTP + Exercise (MC), MPTP + Exercise + Yeast (MC) [MPTP + Exercise + Milmed® (MC)], MPTP + Exercise (SC), and MPTP + Exercise + Yeast (SC).

Pairwise testing using *Tukey's HSD* test indicated the following differences: The MPTP group that received the exercise—Milmed® combination, i.e., MPTP + Exercise + Yeast

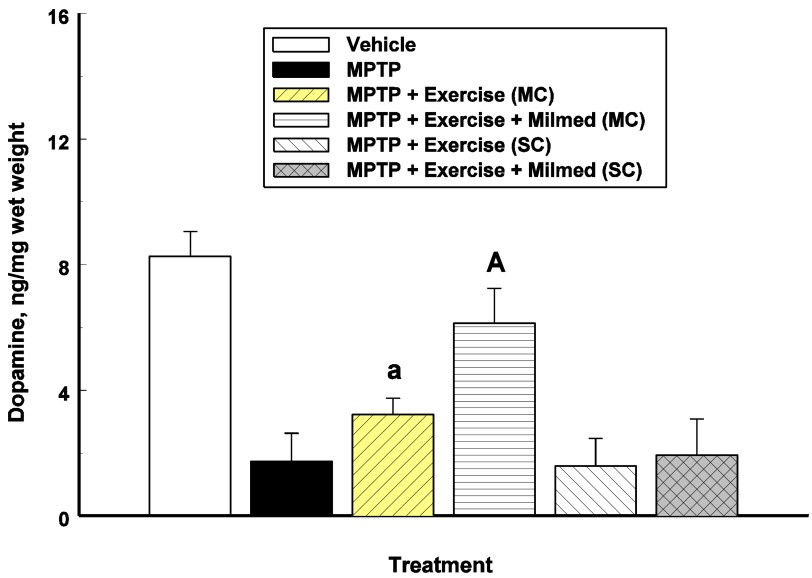

**Figure 3 Neurochemical analysis.** Mean (SD) dopamine concentrations for each of the six groups: Vehicle, MPTP, MPTP + Exercise (MC), MPTP + Exercise + Yeast (MC) [MPTP + Exercise + Milmed® (MC)], MPTP + Exercise (SC), and MPTP + Exercise + Yeast (SC) [MPTP + Exercise + Milmed® (SC)]. MPTP was injected (40 mg/kg, s.c., single weekly injections) either twice or four times prior to initiation of wheel-running exercise (30-min sessions/week, Mon.–Thurs.) + Milmed treatment (four injections, p.o., each week). Pairwise comparisons: A versus MPTP group, $p < 0.01$, a versus MPTP group, $p < 0.05$.

(MC), showed higher dopamine concentrations than the MPTP + Exercise (MC) [MPTP + Exercise + Milmed® (MC)] group which in turn showed higher dopamine concentrations than the MPTP, MPTP + Exercise (SC), and MPTP + Exercise + Yeast (SC) [MPTP + Exercise + Milmed® (SC)] groups. Expressed as percent of control (Vehicle) values, the following were obtained: MPTP = 20%; MPTP + Exercise (MC) = 40%; MPTP + Exercise + Yeast (MC) [MPTP + Exercise + Milmed® (MC)] = 76%; MPTP + Exercise (SC) = 19%; and MPTP + Exercise + Yeast (SC) [MPTP + Exercise + Milmed® (SC)] = 23%.

## DISCUSSION

The purpose of this study was to ascertain whether or not the combination of exercise with Milmed® treatment would restore MPTP-induced functional and neurochemical deficits. The results showed that wheel-running exercise (30-min sessions, 4 days/week) combined with the treated yeast Milmed® suspension (administered 4 times/week), the MPTP + Exercise + Yeast (MC) [MPTP + Exercise + Milmed® (MC)] group, restored spontaneous motor activity markedly by test day 10, restored completely subthreshold L-Dopa-induced activity, and dopamine concentration to 76% of control values, in the condition wherein two administrations of MPTP (2 × 40 mg/kg) were given prior to initiation of exercise and/or Milmed® treatment. Physical exercise by

itself, MPTP + Exercise (MC) group, attenuated these deficits only partially, as has been observed several times previously (*Archer & Fredriksson, 2010*; *Archer & Fredriksson, 2012*; *Archer & Fredriksson, 2013a*; *Archer & Fredriksson, 2013b*; *Archer & Fredriksson, 2013c*; *Fredriksson et al., 2011*). Administration of 4 injections of MPTP each week ($4 \times 40$ mg/kg) induced deficits that were far too severe for amelioration by exercise and Milmed®: i.e., groups MPTP + Exercise (SC) and MPTP + Exercise + Yeast (SC) whereas the MPTP group received no exercise access.

Throughout the published series of experiments (*Archer & Fredriksson, 2010*; *Archer & Fredriksson, 2012*; *Archer & Fredriksson, 2013a*; *Archer & Fredriksson, 2013b*; *Archer & Fredriksson, 2013c*; *Fredriksson et al., 2011*) and (T Archer, 2014, unpublished data), applying different MPTP dose regimes and number of administrations, the percentage increase in striatal dopamine levels, following the exercise invention, has varied as follows: 15% (5 weeks of exercise), 47% (14 weeks of exercise), 44% (7 weeks of exercise), 21% (14 weeks of exercise), 20% (10 weeks of exercise), 42% (14 weeks of exercise), 27% (10 weeks of exercise) and in the present experiment 20% (10 weeks of exercise). Despite this consistent evidence that running-wheel exercise induced reliable elevations in striatal dopamine concentration, it is obvious that exercise by itself was not sufficient to ensure complete restoration. Nevertheless, for the integrity of dopamine neurons, physical exercise throughout exerted an essential and central role: "use it or lose it". Combining running-wheel exercise with Milmed® administration induced complete restoration of striatal dopamine concentrations (*Archer & Fredriksson, 2013a*; *Archer & Fredriksson, 2013b*; *Archer & Fredriksson, 2013c*). In the present study, the treatment with exercise + Milmed® induced a striatal dopamine level that was 76% of the control value, or a percentage increase of 56% over the 10 weeks of the treatment combination. However, it must be considered that prior to the treatment intervention a total of 80 (40 + 40) mg/kg of MPTP neurotoxin had been administered, after introduction of the treatment intervention, a further 80 (40 + 40) mg/kg MPTP was administered. In the *Archer & Fredriksson (2013b)* and *Archer & Fredriksson (2013c)* studies, the 1st two weeks of exercise + Milmed® treatment combination were incorporated prior to the 1st administration of MPTP whereas in the *Archer & Fredriksson (2013a)* study, a $3 \times 30$ mg/kg dose regime of MPTP was applied and the exercise + Milmed® treatment combination was introduced after the 1st administration of MPTP. Thus, the MPTP dose regimes administered in all those studies was substantially milder that employed in the present experiment; indeed, $2 \times 40$ mg/kg of MPTP induces a substantial lesion (*Archer & Fredriksson, 2003*; *Archer & Fredriksson, 2006*), whether followed by a further $2 \times 40$ mg/kg of the neurotoxin or not. Throughout the series of experiments applying the physical exercise + Milmed® interventions to ameliorate or restore the loss of dopamine, it has been indicated that both this intervention, and that of physical exercise by itself, has generated neuroreparative and neurogenesis processes, likely mediated through brain-derived neurotrophic factor (BDNF) (*Archer & Fredriksson, 2013b*).

The clinical implications of physical exercise for improving the patients' condition in Parkinsonism are multiple: e.g., progressive high-intensity locomotor training with body weight support improved their clinical status, quality-of-life and gait capacity as well as

being practicable and well-tolerated (*Rose et al., 2013*). A program of 12-week walking both for Parkinson's disease patients and community-dwelling older adults was shown to be effective: it was found that there were velocity and step-length in the Parkinson's disease group (*Cheng et al., 2013*). In a review of implications for rehabilitation, *Eriksson, Arne & Ahlgren (2013)* have forwarded the notion that physical exercise constitutes an essential ingredient in the process of retaining the healthy self in older individuals with Parkinson's disease. Since L-Dopa remains the drug-of-choice in treatment of Parkinson's disease, it is important to observe that the exercise + Milmed® combination restored completely motor activity after administration of the subthreshold dose (5 mg/kg) of L-Dopa (see Fig. 2). Nevertheless, the emergence of side effects with L-Dopa remains a continual hazard (*Cerasa et al., 2014*; *Pietracupa et al., 2014*; *Shin et al., 2014*). However, it has been shown also in 6-hydroxydopamine-injected rats, an animal model of Parkinson's disease, that L-DOPA-induced dyskinesias were attenuated through the intervention with an exercise schedule (*Aguiar et al., 2013*). *Grazina & Massano (2013)* have presented three conclusions in conjunction with the putative influences of physical exercise upon the symptoms expressions and prognosis in Parkinson's disease: (i) exercise is associated with a lower propensity for developing Parkinson's disease symptoms, (ii) it has been demonstrated that exercise ameliorates, but does not eliminate, disease symptoms, mobility loss, balance problems, gait instability and lesser quality of life (it appears that walking training, tai-chi and tango dancing have demonstrated the highest level of evidence of efficacy); and (iii) that neuroprotective effects accumulating from elevated neuroplasticity may be expected in Parkinsonism conditions, despite the occurrence of these observation from animal studies exclusively. The present findings, taken together with previous observations (*Archer & Fredriksson, 2013a*; *Archer & Fredriksson, 2013b*; *Archer & Fredriksson, 2013c*), both underline these benefits and implicate the role of Milmed® combined with physical exercise to produce more dramatic manifestations of reclaimed dopamine-integrity following disorder onset.

In summary, the lesioning effects of MPTP upon dopamine neurons were introduced either twice or four times before access to running-wheel exercise and/or administration of the treated yeast, Milmed®. In the former condition, the co-treatment of exercise + Milmed® restored both functional, motor activity, and neurochemical, dopamine levels, integrity to a notable extent. Exercise, by itself, attenuated the motor activity deficit and loss of dopamine. In the latter condition, the administration of 4 doses of MPTP (40 mg/kg), a total of 160 mg/kg induced an extent of tissue destruction that proved to be far too severe for later exercise + Milmed® intervention to affect. As we, and others, have described previously (*Archer & Fredriksson, 2013a*; *Archer & Fredriksson, 2013b*), exercise by itself mobilizes neurogenesis and neuroreparative processes in brain regions that have suffered insult; in the present study, the restorative effects upon motor function and dopamine integrity, with particular efficacy in combination with Milmed®, have expressed this propensity of this intervention.

### Funding

This research was supported by a grant from Milmed® AB to Uppsala University. The funders had no role in study design, data collection and analysis, decision to publish, or preparation of the manuscript.

### Grant Disclosures

The following grant information was disclosed by the authors:
Milmed® AB.

### Competing Interests

The authors declare that there are no competing interests.

### Author Contributions

- Trevor Archer conceived and designed the experiments, analyzed the data, contributed reagents/materials/analysis tools, wrote the paper, reviewed drafts of the paper.
- Danilo Garcia wrote the paper, reviewed drafts of the paper.
- Anders Fredriksson conceived and designed the experiments, performed the experiments, analyzed the data, contributed reagents/materials/analysis tools, wrote the paper, prepared figures and/or tables, reviewed drafts of the paper.

### Patent Disclosures

The following patent dependencies were disclosed by the authors:
Milmed European patent number (Trevor Archer, Anders Fredricksson), EP 2470213

### Animal Ethics

The following information was supplied relating to ethical approvals (i.e., approving body and any reference numbers):

The study was carried out in accordance with the European Communities Council Directive of 24th November 1986 (86/609/EEC) after approval from the local ethical Committee (Uppsala University and Agricultural Research Council), and by the Swedish Committee for Ethical Experiments on Laboratory animals (License S93/92 and S77/94, Stockholm, Sweden).

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
