# Peer review of "Restoration of MPTP-induced deficits by exercise and Milmed® co-treatment"

_PeerJ, doi:10.7717/peerj.531_

## Round 0.1 · original submission · Minor Revisions

Dear Authors.Please revised the manuscript as per suggestions of the peer reviewers.The revised manuscript will be re-reviewed by the same persons therefore please revise the manuscript carefully.

Reviewer 1 ·

Basic reporting

The manuscript is in very short presentation, results and discussion.
I do not see how they can contribute to the advancement of knowledge in the area.
I suggest the rejection of the article.

Experimental design

The manuscript is in very short presentation, results and discussion.
I do not see how they can contribute to the advancement of knowledge in the area.
I suggest the rejection of the article.

Validity of the findings

The manuscript is in very short presentation, results and discussion.
I do not see how they can contribute to the advancement of knowledge in the area.
I suggest the rejection of the article.

Additional comments

The manuscript is in very short presentation, results and discussion.
I do not see how they can contribute to the advancement of knowledge in the area.
I suggest the rejection of the article.

Reviewer 2 ·

Basic reporting

No Comments

Experimental design

No Comments

Validity of the findings

No Comments

Additional comments

The author concluded that MPTP induced dopamine related behavior deficits ameliorated by exercise and treatment with Milmed yeast in moderate condition, but not in severe condition. Although, MPTP and their effects have been well studied in different Parkinson disease animal models, this work would be appreciable. However, to strength the present work, need to clarify some questions and must include in discussion part,

How dose Milmed yeast stimulate the locomotion activity as well as dopamine level?. Is there any kind of pathway involved
or is there any previous reference reported dopamine and Milmed pathways?.

Since, Milmed shows beneficial role in PD associated diseases, good to know that after oral administration of Milmed yeast,
Is there any intestinal infection found in mice? Or any side effect if administrated with high doses?

Besides, has to be corrected in below mentioned comments,

In abstract, Better to start with a few lines of introduction. Line 2, expand MPTP. C57BL6 instead of C57BL/6, correct everywhere in text. Line 4, spelling mistake; yimes/week. Line 13, far has to be removed.

In Introduction, Line 9, expand MPTP. Line 28, expand PD. Line 47, instead of (moderate condition (MPTP +Exercise+Milmed (2), better to change like (moderate condition (MC) (MPTP+Exercise+Milmed (MC) like same in severe condition (SC) (MPTP+Exercise+Milmed (SC). Numbers make readers confusion. Also correct the same in results and discussion parts, it has to be changed.

In Methods and Material, How many animals per groups? Line 57, change C57BL/6, correct five-to –a sentence not completed . Line 58, 22 ± 1oC, 12 h/12 H to be changed . Line 75, need expansion of ADEA testing chamber of motor activity. Line 88, change word scarified instead of killed .

In Results It would be better to present the data Mean ±SEM instead of SD. Change in all result parts. Change instead of F(41, 419) = 97.61 to F(41, 419) = 97.61. change in all result parts. Line 118, 3 to days, something is missing,

In Discussion, Line 157, check alignment, Line 191, remove that

In Figures, In Figures, a, A, B and b has been mentioned but it is difficult to understand. Instead these letters, better to use symbols. Also describe in legends, which one is compared with those groups. Graphs are not clear; better to use good resolution using Photoshop.

·

Basic reporting

No Comments

Experimental design

No Comments

Validity of the findings

No Comments

Additional comments

This manuscript reports a neurobehavioral study of restoration of MPTP-induced deficits by Exercise and Milmed Co-treatment using mice behavioral and neurochemistry experiments about the combination of physical exercise with Milmed treatment. These results are so innovative, valuable and new findings in the study, furthermore, before consideration for publication, some points should be addressed.

Specific points:
- The number of C57 BL6 mice in each group has not been mentioned, and this may be important.
- In the study, MPTP+Exercise+Yeast(2) group showed higher DA concentrations than other groups, can the authors explain briefly the mechanisms how MPTP+Exercise+Yeast(2) group induces markedly the release of DA?
Minor points:
- In the abstract (pag.1 line4), the spelling mistaken, ‘yimes instead of times’.
- In the abstract (pag.1 fourth line from bottom), ‘MPTP+Exercise+Yeast(2) instead of MPTP+Exercise+Yeast(4)’.
- In the figure1 (pag.14 line3), ‘MPTP+Exercise+Yeast(4) instead of MPTP+Exercise+Yeast(2)’.
- In the figure2 (pag.16 line3), ‘MPTP+Exercise+Yeast(4) instead of MPTP+Exercise+Yeast(2)’.
- In figures, all p-values are presented as alphabet, but there was no explained in the legends.

·

Basic reporting

See below

Experimental design

See below

Validity of the findings

See below

Additional comments

Recommended for publication with minor corrections. The subject matter is relevant and the work done appears to be scientifically sound. Please revise writing to improve grammatical and typo errors, including those found in the abstract.

It'd also be good to enhance the quality and clarity of the 3 figures.

Also the reference list needs to be polished up.

---

## Round 0.2 · accepted · Accept

Dear Authors,Thank you for your submission of this manuscript which has been accepted for publication.Congratulations.

Reviewer 2 ·

Basic reporting

No comments

Experimental design

No comments

Validity of the findings

No comments

Additional comments

No comments

·

Basic reporting

No Comments, well written

Experimental design

No Comments, well designed study

Validity of the findings

This revised manuscript reports a neurobehavioral study of Parkinson`s diseasae using mice behavioral and neurochemistry experiments. These results are valuable and new findings in both motor functional and dopamine chemistry in the study.

Additional comments

I believe the manuscript to be interesting and insightful, and particularly well written. This study is well designed and explained about the combination of physical exercise with Milmed treatment on Parkinson`s disease. In the study, the MPTP+Exercise+Yeast(MC) group showed that spontaneous motor activity restored markedly by test day 10, and increased the level of dopamine concentration. The Parkinson`s disease is a slowly progressive neurodegenerative disease so that the time point of disease phase needs to be further investigated.

·

Basic reporting

Acceptable as corrections have been made.

Experimental design

Acceptable.

Validity of the findings

Acceptable.

Additional comments

Please correct spelling for Physiology in the Author's affiliation.